# Chemical Profiles, In Vitro Antioxidant and Antifungal Activity of Four Different *Lavandula angustifolia* L. EOs

**DOI:** 10.3390/molecules28010392

**Published:** 2023-01-02

**Authors:** Claudio Caprari, Francesca Fantasma, Pamela Monaco, Fabio Divino, Maria Iorizzi, Giancarlo Ranalli, Fausto Fasano, Gabriella Saviano

**Affiliations:** Department of Bioscience and Territory, University of Molise, C.da Fonte Lappone snc, 86090 Pesche, Italy

**Keywords:** *Lavandula angustifolia* L., essential oil, GC-MS analysis, scavenging activity, antifungal activity, biodeteriogen control

## Abstract

*Lavandula angustifolia* L., known as lavender, is an economically important Lamiaceae due to the production of essential oils (EOs) for the food, cosmetic, pharmaceutical and medical industries. The purpose of this study was to determine the chemical composition of EOs isolated from four inflorescences of *L. angustifolia* L. collected in different geographical areas: central-southern Italy (LaCC, LaPE, LaPS) and southern France (LaPRV). The essential oils, obtained by steam distillation from plants at the full flowering stage, were analyzed using gas chromatography coupled with mass spectrometry (GC-MS). More than 70 components identified in each sample showed significant variability among the main constituents. The four EOs analyzed contained the following as main component: linalool (from 30.02% to 39.73%), borneol (13.65% in LaPE and 16.83% in La PS), linalyl acetate (24.34% in LaCC and 31.07% in LaPRV). The EOs were also evaluated for their in vitro antifungal activity against two white rot fungi (*Phanerochaete chrysosporium* and *Trametes cingulata*) as potential natural biodeteriogens in the artworks field, and against *Sclerotium rolfsii*, *Botrytis cinerea* and *Fusarium verticilloides* responsible for significant crop yield losses in tropical and subtropical areas. The results confirm a concentration-dependent toxicity pattern, where the fungal species show different sensitivity to the four EOs. The in vitro antioxidant activity by DPPH assay showed better scavenging activity on LaCC (IC_50_ 26.26 mg/mL) and LaPRV (IC_50_ 33.53 mg/mL), followed by LaPE (IC_50_ 48.00 mg/mL) and LaPS (IC_50_ 49.63 mg/mL). The potential application of EOs as a green method to control biodeterioration phenomena on a work of art on wood timber dated 1876 was evaluated.

## 1. Introduction

Essential oils (EOs) are accumulated in different organs and/or structures of aromatic plants, i.e., fruits, flowers, leaves, seeds, barks, and their components are being classified as secondary metabolites. They are relatively small chemical compounds, with ubiquitous distribution in the plant kingdom, though their role in the life of plants in most cases is not known [1]. Their wide range of activity is determined both by the plant genotype that determines its chemical composition and by other factors such as environmental condition, geographical location, time of harvest, stage of plant development or extraction methodology [2,3,4].

In recent years, the application of plant extracts and plant EOs has received increasing interest as a natural alternative to the use of commercial synthetic chemicals to control the main postharvest diseases of fruits and vegetables. As an example, different EOs such as oregano, winter savory, eucalyptus and peppermint, along with many of their principal components (limonene, carvacrol, etc.), have already demonstrated attractive and important antimicrobial, insecticidal, antioxidant and herbicidal activity for the agri-food industry [5]. Furthermore, EOs are biodegradable, have minimal effects on non-target organisms and slow down the emergence of drug resistance in parasites [6]. These EO features are commercially popular and encourage their potential use as a natural treatment to prevent and reduce the spoilage of crops and food and extend their shelf life, as well as to control weeds [7,8].

The *Lavandula* genus (Lamiaceae family), a small fragrant shrub native to the Mediterranean area, is cultivated worldwide for its EOs with excellent aroma, which find application in various industries, such as perfumery, pharmaceuticals and cosmetics. Lavender EO is known to possess sedative, carminative, anti-depressive and anti –inflammatory properties [9], and it is effective against the growth of a wide range of microorganisms and has antioxidant activity [10]. Although the *Lavandula* genus is one of the most well-known EO crops in the world, with its 40 species, numerous hybrids and about 400 officially registered cultivars [11,12], only three of these species have high commercial value. The principal cultivated species for aromatic oils are fine lavender (*Lavandula angustifolia* L.), spike lavender (*Lavandula latifolia*) and lavandin *(Lavandula intermedia*), a sterile hybrid of *L. angustifolia* x *L. latifolia* [13].

The industrial cultivation and production of *L. angustifolia* L. and *L. intermedia* as medicinal and aromatic plants has increased rapidly in recent years, and worldwide interest in *Lavandula* EOs is still increasing. The EO market of lavender was USD 530.5 million in 2020 and is expected to reach USD 864.7 million by 2025, with a compound annual growth rate (CAGR) of 10.3% from 2020 to 2025 [14]. Lavender species EOs have the same chemical composition, but differ in the proportions of their components and in the yields after extraction. Common criteria for the determination of oil quality are camphor, linalool and linalyl acetate levels [15]. For instance, lavender EO contains linalyl acetate (25–45%), linalool (25–38%) and camphor (0.5–1.0%), while lavandin EO contains linalyl acetate (28–38%), linalool (24–35%) and camphor (6–8%), respectively, according to ISO 8902:2009 [16]. Data from the literature show that more than 100 components have been identified; the main active ingredients in *L. angustifolia* flowers’ EO are monoterpenes (linalool, linalyl acetate, lavandulol, geraniol, bornyl acetate, borneol, terpineol, eucalyptol (also known as 1,8-cineole) and lavandulyl acetate) in different percentages, according to the geographic area of origin [17]. In particular, lavender EO from Italian regions shows a high concentration of linalool (ranging from 35% to 36%), linalyl acetate (2.75% to 21%), camphor (5 to 11%), 1,8-cineole (3 to 10%) and borneol (2% to 19%) [18,19,20].

It is worth noting that EOs carry out an antimicrobial action in different ways: by altering the membrane structures and cell walls or by modifying the metabolism by modifying/blocking the enzymatic reactions or specific enzymes [21,22].

Currently, numerous studies show that natural compounds represent a valid alternative to traditional biocides, which are generally toxic and non-degradable, that can persist for a long time in the environment [23,24,25,26]. White rot fungi cause white rot on wood or trees and belong to the Basidiomycete family. The mycelium penetrates the cell cavity and releases ligninolytic enzymes that decompose the wood into a whitish sponge-like mass. Thanks to the cellulose fibrils, wood retains its elasticity but is more fragile than uninfected tissue. The cellulose fibrils can be last degraded or only partially altered [26].

Although they effectively break down lignin, these fungi cannot utilize it as an energy source and are assumed to degrade it for access to cellulose in the cell wall [27]. White rot fungi exhibit this unique ability, which lends them great potential interest in the pretreatment of biomass in many biotechnological applications, such as the production and treatment of wood pulp and its bleaching and biofuel production [28]. On the other hand, in the field of biorestoration, they can pose a challenge in terms of contrasting the superficial and/or deep biodeteriogen colonization on wooden and stonework surfaces [29].

In the literature, there are many data on the chemical composition and biological activity of EOs extracted from many species, but despite that, there is a lack of comparative studies on the biological activity of EOs of *L. angustifolia* L. as food preservatives and against postharvest fruit pathogens [30,31].

The present study aimed to investigate the efficacy of EOs of *L. angustifolia* on different fungal species involved in both plant pathogen interaction and wood biodeterioration. Specifically, the following steps were performed: (i) comparison of the chemical composition of the EOs of four lavender flowers from different geographic regions; (ii) statistical analyses of data; (iii) in vitro antifungal activities against five phytopathogenic fungi; (iv) in vitro antioxidant activities of the EOs by DPPH free radical scavenging assay; (v) case study, as green biocidal (against biodeteriogen) on altered wooden artwork.

## 2. Results

### 2.1. EO Yield and Composition

Inflorescence hydrodistillation on four populations of *L. angustifolia* L., harvested in Rosciano (PE) (central Italy, LaPe), Pesche (IS) (south Italy, LaPS), Capracotta (IS) (south Italy, LaCC), and Forcalquier in Provence (France, LaPRV), was performed.

All extracts provided an essential oil characterized by a typical smell in a yield ranging between 3.1 and 5.9%, calculated according to the initial weight of 100 g each, respectively. The characterization of all EOs was evaluated using GC/MS and a set of standards: linalool, borneol, terpinen-4-ol, camphor and lavender oil (Sigma Aldrich, St. Louis, MO, USA).

Table 1 summarizes the chemical composition of the EOs and their experimental retention indices compared with the retention indices reported in the literature [32], their percentage compositions and the abbreviations of the different classes of terpenes; the compounds are reported according to their elution on a Rtx^®^-5 Restek capillary column. Approximately 70 components were identified for each sample, showing significant variability in some major constituents.

Furthermore, a statistical analysis using the ANOVA test and the post hoc Duncan test was performed on the data in Table 1. In particular, for each compound, values of percentages that are significantly different from each other were identified; the results are reported in Appendix A.

GC-MS analysis confirms the presence of linalool, between 30 and 40% of the total composition, as the major constituent and characterizing essences of *L. angustifolia* L. In contrast, among the other constituents characterizing lavender EOs, marked differences are evident (Table 1). It is known that a variability in the chemical composition of EOs may induce great variability in their biological activity against microorganisms. Environmental factors (soil composition, temperature, altitude, climate, etc.) are among the factors conditioning the variability of the metabolic profiles detected in this study [4]. In this contest, eucalyptol (1,8 cineole), camphor, borneol, terpinen-4-ol, lavandulyl acetate, (E)-caryophyllene and, overall, linalyl acetate present major variability in percentage when compared with each other. Total ion chromatograms of the EOs analyzed are displayed in Figure 1. Oxygenated monoterpenes represent the most abundant class for all EOs examined (78%, 79%, 81% and 86%), followed by monoterpenes (8%, 13%, 12% and 5.8%), sesquitepenes (5.5%, 2.6%, 2% and 1.8%) and oxygenated sesquiterpenes (3.2%, 1.9%, 1% and 5.2%), as shown in Table 2. Major oxygenated monoterpenes included aliphatic AMO (linalool 30 ÷ 40%, and linalyl acetate 1.8 ÷ 31%), bicyclic monoterpenes BMO (1,8-cineole (or eucalyptol) 1.2 ÷ 13.5%, camphor 0.5 ÷ 6%, borneol 1 ÷ 17%) and monocyclic monoterpene MMO (terpinen-4-ol 1.6 ÷ 10%) in different amounts for each lavender EO.

### 2.2. Explorative Data Analysis

In order to characterize the levels of diversity in the chemical composition of EOs, the classic *Shannon* entropy [33] has been considered with its relative version, the *Pielou* index [33]. Furthermore, to compare compositions between samples and to assess the levels of dissimilarities between compositions of different types of lavender EOs, the *percent model affinity* (*PMA*) index has been calculated [34]. Mathematical details are presented in Section 4.5.

The results are shown in Table 3, where the reported mean and standard error of each indicator are calculated over the three sampling replicates. In addition, to better show which compounds characterize the levels of dissimilarities between types of lavender EOs, the cross plots between compositions are reported in Figure 2. The blue line is the bisector; it represents the situation in which a certain compound is present in the same percentages in both types of lavenders considered in each panel (Figure 2a–f). The points that are significantly distant from that bisector represent compounds observed in rather different percentages; they are highlighted by the specific names. Here, a certain point is considered distant from the bisector when its distance is larger than the standard deviation *S_d_* = 2.12. The dashed lines represent the tolerance interval, bisector ± *S_d_*. The points within this interval represent compounds with approximately the same percentage in both types of lavender EOs.

The results in Table 3 clearly show that two pairs of similarities are observed, LaPE with LaPS (PMA = 0.866) and LaCC with LaPRV (PMA = 0.719), while in terms of diversity, all the types of lavender Eos present the same level of entropy, suggesting a homogeneous structure in the chemical composition for all four types of Eos.

### 2.3. Antifungal In Vitro Test

The antifungal activity of the four lavender EOs against the growth of *S. rolfsii*, *B. cinerea*, *F. verticillioides*, *P. chrysosporium* and *T. cingulata* phytopathogenic fungi was carried out in vitro. LaCC-, LaPE-, LaPS- and LaPRV-characterized lavender EOs were used at different concentrations, ranging from 0 to 40 µL, as described in Section 4.6.

Figure 3 reports the reduction in fungal growths, expressed as a reduction in mycelium radial growths on Petri dishes with respect to the negative control.

The results show, first of all, a concentration-dependent trend in antifungal activity; then, each fungus shows a different profile of interaction and a different sensitivity among the tested EOs. Total inhibitory growth is generally reached using an amount ranging from 20 to 40 µL. *F. verticillioides* (Figure 3e) shows slightly different radial growth curves; in this fungus, EOs have a lower performance in terms of fungal growth inhibition. For example, on *F. verticillioides,* LaPE is less effective, with about 50% growth inhibition with 40 µL of the analyzed sample. On the contrary, LaPE applied on *P. chrysosporium* (Figure 3a) shows better inhibitory activity. In the same experimental conditions, it inhibits the growth of *P. chrysosporium* mycelium by 100% with only 20 µL of EO. Good inhibitory activity was found on *S. rolfsii* and *B. cinerea* using 30 µL of LaPS (Figure 3b,c), while the same amount of LaCC showed a better inhibition growth kinetic on *T. cingulata* (Figure 3d). LaPRV EO showed the best inhibition growth kinetics on F. verticillioides, although this fungus proved to be the least sensitive to treatment with EOs. The positive controls for antifungal activity were carried out using PDA plates added with Thiram (Tetrasar 50, powder, Isagro srl) at final concentrations in the ranges of 0 and 73 µg/mL (Figure 3f). *B. cinerea* and *P. chrysosporium* show linear and total mycelial growth inhibition at 73 µg/mL of Thiram. *T. cingulata* is 10 times more sensitive to Thiram, while *F. verticillioides* and *S. rolfsii* need a higher dose of fungicide to stop mycelial growth. In particular, the *S. rolfsii* toxicity curve flattens at higher doses of fungicide, indicating a lower sensitivity of the fungus to high doses of Thiram.

### 2.4. Antioxidant Assay

The examined EOs of *L. angustifolia* L. showed different antioxidant activity, expressed in terms of IC_50_. The LaCC sample exhibited the highest antioxidant capability with an IC_50_ value of 26.26 ± 0.21 mg/mL, followed by the LaPRV sample (IC_50_ = 33.53 ± 0.23 mg/mL) (Table 4). As shown in Figure 4, LaPE and LaPS EOs had comparable scavenging activity, with IC_50_ values of 48.00 ± 0.35 and 49.63 ± 0.42 mg/mL, respectively. The scavenging activity ranged from 13.08% to 98.70% on LaPS and LaCC EOs, using concentrations ranging between 8.71 mg/mL and 261.21 mg/mL, respectively. The IC_50_ value of standard ascorbic acid was 0.024 + 0.004 mg/mL.

### 2.5. Case Study

#### 2.5.1. Historical Work of Art: Timber of Wood Dated 1876

The old wood appeared to be in an altered condition. A large portion of the scratched surface of the wood was affected by material gaps and traces of light white-gray residues of a nature yet to be ascertained (Figure 5). The wood showed visible signs of degradation, with irregular loss of material and texture. White powdery residues were not visible on the opposite side of the wood, indicating that the wood may have been used less or that more care was taken when cleaning it after use.

#### 2.5.2. Microbial Counts–Growth on Sample Surfaces Was Analyzed

The microbial growth that was detected revealed the sole presence of fungi and an absence of bacteria in almost all samples analyzed. Higher values for fungal counts were observed on the altered wooden timber, on the larger side surface with average values of <10 CFU/cm^2^. The microbial diversity in the analyzed samples was very low and revealed the presence of one predominant fungal type; this may be due to the peculiar indoor environmental conditions, such as those present in the confined room of the Mostra Permanente (Museum). The inocula were smeared on the surface of Petri dishes containing malt agar 2% (Difco) added with streptomycin sulfate (Sigma-Aldrich, St. Louis, MO, USA) and 100 µg/mL ampicillin (Fisher BioReagents, Monza, Italy). The plates were incubated at 26 °C for 4 days. The fungal isolate was identified as *S. rolfsii* and it was used together with four other plant pathogenic fungi to evaluate the antifungal activity of lavender EOs (see Section 2.3).

## 3. Discussion

Fungal infections are not just a human health problem, they also affect the fields of agriculture and cultural heritage [36,37,38]. It is estimated that 20% and 40% of the total agricultural productivity loss is caused by animals, weeds and pathogens. These losses have implications for human health, the environment, and the economy [37]. To cite some data, in the 21st century, it is estimated that the loss of crops is due to 18% from animal parasites and 16% from microbial diseases (a large majority due to phytopathogenic fungi), for an average loss of 68% of the tonnage of agricultural production [39].

Microorganisms (lichens, algae, fungi and bacteria) are biodeteriogens and agents of colonization on artwork surface [29].

Plant extracts and EOs are ecological, protective, curative and antagonistic to many diseases. Therefore, plant extracts may have an important role in controlling soil-borne diseases, as they are a rich source of bioactive substances [40]. For example, thyme (*Thymus vulgaris*) EO is already known to be effective against fungi due to its high concentration of thymol and carvacrol [41].

Known in aromatherapy for its relaxing and sedative virtues, lavender EOs are evaluated in this study for their effectiveness against microorganisms, including fungi [42]. A comparison among the chemical components of four lavender EOs reveals linalool, linalyl acetate and borneol as the most representative compounds. However, differences occurred among samples, suggesting that a different chemical composition is subject to change under the influence of several aspects, such as climatic conditions and environmental factors. This resulted in variability in antifungal and antioxidant activity. The percent model affinity (PMA) index (Table 3) was used to identify similarities between the four lavender EOs. Two pairs of lavender EOs, LaPE/LaPS (PMA = 0.866) and LaCC/LaPRV (PMA = 0.719), were identified as having a high similarity level, while, in terms of “diversity” (Table 3), homogeneous structure in the chemical compositions was suggested for all the four types of the EOs. In the LaPE/LaPS pair, along with linalool, we observed a high level of borneol (13.65% and 16.83%), limonene (3.83% and 3.43%), camphor (5.68% and 3.87) and terpinen-4-ol (8.2% and 9.98%), while the concentration of linalyl acetate (1.8% and 2.41%) and lavandulyl acetate (0.49% and 0.98%) was very low (Table 1). In the LaCC/LaPRV pair, the representative components were linalyl acetate (24.34% and 31.07%) and lavandulyl acetate (6.51% and 3.2%), while borneol (1.05% and 4.43%) and limonene (0.25% and 0.75%) were in low concentrations.

A relatively small amount of linalyl acetate in comparison to the literature data has already been found in earlier studies [20]. The cross plots of Figure 2 reveals other components, particularly monoterpenoids (1,8 cineole, terpinen 4-ol and camphor), that contributed to the diversity among the lavender pairs. Most of these compounds belong to the alcohol group and, together with linalool, represent the monoterpenoid component that was the most abundant in all four lavenders (Table 2). In our analysis, we also observed that the composition of the EO from LaCC was more complex than that of the other samples; *E*-*β*-ocimene (3.45%), terpinen-4-ol (5.42%), *E*-caryophyllene (2.78%), caryophyllene oxide (2.84%) and lavandulol (1.98%) were also detected. This feature could explain the better antioxidant activity observed with respect to the other EOs.

Several methods are used to evaluate the antioxidant activity of EOs obtained from different plants; however, differences in these methods may lead to different results that make comparisons difficult, and thus, investigations on the modification and improvement of these methods still continue to provide the most reliable technique [43]. We chose the DPPH method because it is a popular, quick, easy and convenient approach for the measurement of antioxidant properties involving the use of free radicals to assess the potential of substances to act as hydrogen donors or free radical scavengers [44,45]. Moreover, several parameters affect EO composition that may also result in different antioxidant activity values [46,47].

Furthermore, the results of the antioxidant activities of the four lavenders can be explained by considering the similarity between the same pairs (LaCC/LaPRV and LaPE/LaPS). Table 4 reports the IC_50_ values (mg/mL) and scavenging activity range (%) of the four lavender EOs analyzed. The composition of the EOs may explain the different values of IC_50_ in the antioxidant test. The higher antioxidant activity detected in LaCC EO (IC_50_ 26.26 mg/mL) could be related to the synergistic property associated with the minor components in the mixture, as reported in the literature [48,49,50]. LaCC EOs, along with linalool, linalyl acetate and lavandulyl acetate as the major components, contain several minor terpenoids, such as *E*-caryophyllene, which possess antioxidant, anti-inflammatory and analgesic properties [51]. Ruberto et al. [52] tested about 100 pure compounds present in EO for their antioxidant effectiveness. More recently, antioxidant activity has been shown in linalool, which is the dominant terpenoid in all four EOs tested [53].

Antifungal activity was performed in vitro on two white rot fungi (*P. chrysosporium* and *T. cingulata*) and three others responsible for rotting and diseases in various organs of the plant (*S. rolfsii*, *B. cinerea* and *F. verticillioides)* using four different *L. angustifolia* EOs. Results highlighted that LaPS shows efficacy against *S. rolfsii*, LaPE is more active against *P. chrysosporium*, LaCC inhibits *T. cingulata* and, finally, LaPRV is active against *B. cinerea* and *F. verticillioides*. In all cases, *F. verticillioides* appears to be more resistant to the toxic effect of the EOs used. These results are in accordance with [54,55], which indicate an ED_50_ against *B. cinerea*, *Fusarium* spp. and *Fusarium oxysporum* of 223 μg/mL, 520 μg/mL and 372 μL/mL, respectively. Synergic effects and different cellular targets may be the key to interpretation. While major components have been extensively studied [56,57,58], others minority components play various roles. They may enhance or decrease the synergic effect by modifying the texture, color or density of the oil, but also EO cellular penetration or its lipophilic or hydrophilic nature, its membrane or wall fixation and its distribution within the cell, making the simultaneous inhibition of different cell targets possible [42,59]. In this regard, the lavenders used show a fairly wide range of monoterpenes and sesquiterpenes, between 5.78 and 13.11% and between 1.77 and 5.49%, respectively.

The comparison of data among the lavenders analyzed gives us the opportunity to select one or more populations with distinct EO chemical components with respect to the others. That is, choose a specific lavender carefully on the basis of its chemical composition to expect the results that it offers.

Preliminary results from our experiments could be useful from the perspective of controlling biodeteriogen activity (i.e., fungi) on altered wooden artworks. The indications will allow for the development of an organic green strategy for the recovery of altered works of art, as an alternative to the use of biocides and toxic compounds. The use of *L. angustifolia* EOs as a natural essence, for instance, inside a confined space such as a bag containing an old work of art to be recovered, could be a suitable technical solution [39].

However, not all that comes from nature is safe and free of hazard for human health. Since prehistorical times, the presence of toxic and therapeutic ingredients in natural plants and extracts (roots, leaves, fruits, fungi, etc.) has been well known and has been adopted or excluded in traditional and ethnic medicine around the world. Lavender derivatives, including EO, are not fully free of risk since they are included in the REACH list [60] among the substances that cause “serious eye irritation, are harmful to aquatic life with long-lasting effects and may cause an allergic skin reaction” [61].

The overall results suggest a careful use of each EO of lavender, taking into account its peculiarities, efficacy and limitations for utilization.

## 4. Materials and Methods

### 4.1. Plant Materials

All three Italian *L. angustifolia* flowers were collected in August 2021 in the early hours of the morning and during the balsamic period. The LaPE flowers were harvested at Villa Vanda Farm (42° 20′ 59.83″ N 14° 01′ 54.88″ E) at an altitude of about 150 m and a distance of 20 km from the Adriatic Sea. The flowers are grown in parallel rows alongside a centuries-old olive grove managed according to the rules of organic farming in an area called “Oasi orientale di Villa Badessa,” Rosciano (Pe, Abruzzo Region, Italy). The soil has a predominantly loamy-clayey texture with no summer irrigation (Figure 6a). The LaPS flowers were harvested in the garden of DiBT, Pesche (IS, Molise Region, Italy, 41°36′25″ N 14°15′55″ E), at an altitude of about 700 m, with a medium-textured soil and no summer irrigation. The LaCC flowers were harvested in the Garden of the Apennine Flora of Capracotta (Molise Region, Italy, 41°50′42.06″ N 14°16′35.71″ E), a natural botanical garden that preserves the autochthonous flora of the central-southern Apennines in Italy, at an altitude of about 1550 m. The LaPRV flowers were purchased freshly picked at the Forcalquier market in August 2021 (Provence Region, France, 43°95′97″ N 5°78′8″ E (Figure 6b). All plants were identified at the Department of Bioscience and Territory (University of Molise, Pesche, Italy), and the voucher specimens (LaPe-57-2021; LaPS-58-2021; LaCC-59-2021; LaPRV-60-2021) were deposited in the Herbarium of DiBT, University of Molise. The flowers were immediately used for the hydrodistillation of EOs. The composition of all samples was analyzed and compared.

### 4.2. EOs Isolation

Flowers (100 g) of lavenders were hand-selected, cleaned and then separately subjected to hydrodistillation for 2 h according to the standard procedure described in the European Pharmacopoeia [62]. The EOs were dried over anhydrous sodium sulfate to remove traces of water and then stored in dark vials at 4 °C prior to gas chromatography-mass spectrometry (GC-MS) analysis.

### 4.3. GC-FID Analysis and GC/MS Analysis

The characterization of the EO samples was determined using a gas chromatography system, GC 86.10 Expander (Dani), equipped with a FID detector, Rtx^®^-5 Restek capillary column (30 m × 0.25 mm i.d., 0.25 um film thickness) (diphenyl-dimethyl polysiloxane), a split/splitless injector heated to 250 °C, and a flame ionization detector (FID) heated to 280 °C. The column temperature was maintained at 40 °C for 5 min, and then programmed to increase to 250 °C at a rate of 3 °C/min and held, using an isothermal process, for 10 min. The carrier gas was He (1.0 mL/min); 1 uL of each sample was dissolved in *n*-hexane (1:500) and injected. GC-MS analyses were performed on a Trace GC Ultra (Thermo Fisher Scientific, Waltham, MA, USA) gas chromatography instrument equipped with a Rtx^®^-5 Restek capillary column (30 m × 0.25 mm i.d., 0.25 um film thickness) and coupled with an ion-trap (IT) mass spectrometry (MS) detector Polaris Q (Thermo Fisher Scientific, Waltham, MA). A programmed temperature vaporizer (PTV) injector and a PC with a chromatography station, Xcalibur (Thermo Fisher Scientific, Waltham, MA, USA), was used. The ionization voltage was 70 eV; the source temperature was 250 °C; full scan acquisition in positive chemical ionization was from *m*/*z* 40 up to 400 a.m.u. at 0.43 scan s^−1^. The GC conditions were the same as those described above for the gas chromatography (GC-FID) analysis.

### 4.4. Identification of EO Components

The identification of the essential oil components was based on the comparison of their Kovats retention indices (Exp RI), determined in relation to the tR values of a homologous series of n-alkanes (C8–C20) injected under the same operating conditions as those in the literature [63,64]. The MS fragmentation pattern of each single compound with those from the NIST 02, Adams and Wiley 275 mass spectral libraries was compared [65,66]. The relative contents (%) of the sample components were computed as the average of the GC peak areas obtained in triplicate without any corrections [67]. All analytical standard components utilized (*n*-alcane C8-C20, linalool, borneol, terpinen-4-ol, camphor and lavender oil) were bought from Sigma Aldrich, St. Louis, MO, USA.

### 4.5. Statistical Analysis

The explorative data analysis was performed using the **R** software, available for free under the terms of the Free Software Foundation’s GNU General Public License in source code form [68].

Concerning the results presented in Section 2.2, the levels of diversity in the chemical composition of the EOs were evaluated using the classic *Shannon* entropy:$$H=-\overset{k}{\underset{i=1}{\sum }}p_{i}log\left(p_{i}\right),\;$$
where *p_i_* is the proportion of the *i-th* of *k* compounds observed in the sample, and with its relative version, the *Pielou* index,
$$J=\frac{H}{log\left(k\right)}.$$

Furthermore, the levels of dissimilarities between the compositions of the different types of lavender EOs were assessed using the *percent model affinity* (*PMA*) index,
$$PMA=1-0.5\overset{k}{\underset{i=1}{\sum }}\left|p_{Ai}-p_{Bi}\right|,$$
where *A* and *B* denote two generic samples.

### 4.6. Antifungal Activity Assay

The *S. rolfsii* and four other fungal strains (*B. cinerea*, *F. verticillioides*, *P. chrysosporium* and *T. cingulata*), previously identified and characterized (Figure 7) [69,70,71], were used in this study. Pure essential oils from the samples LaCC, LaPE, LaPS and LaPRV were dissolved in a final volume of 200 µL in ethanol and then added to 19 mL PDA (Oxoid Limited, Basingstoke, Hampshire, UK) plates to obtain the different final concentrations. Mycelial plugs (4 mm in diameter) from the edges of Petri dish cultures were incubated in the center of each PDA plate (90 mm diameter). Fungal cultures were incubated in the dark at 26 °C and 70% relative humidity (RH) for a variable number of days ranging from 3 to 12, depending on the fungal species analyzed (3 days for *P. chrysosporium*, 4 days for *S. rolfsii* and *B. cinerea*, and 12 days for *T. cingulata* and *F. verticillioides*). The tests were conducted in triplicate. The antifungal activity was determined by measuring the diameter (in mm) of the radial growth. The control growth was carried out on PDA plates prepared as described above, but without the EO samples. The positive controls for antifungal activity were carried out using PDA plates added with Thiram (Tetrasar 50, powder, Isagro Srl, Aprilia, Italy) at final concentrations in the range of 0–73 µg/mL.

### 4.7. Antioxidant Activity

Antioxidant activity was determined by assessing the scavenging capacity of antioxidant compounds towards 2,2-diphenyl-1-picrylhydrazyl (DPPH) radical, using the standard method [72] adopted with suitable modifications [73]. In particular, for each EO, different aliquots were added to 1 mL of a freshly prepared DPPH methanolic solution (27 μg/mL) to obtain diverse EO concentrations (as reported in Figure 4). The samples were incubated in the dark at room temperature for 30 min. The absorbance (A) of each sample was measured using a UV-Vis spectrophotometer (Shimadzu UV-1601) at a wavelength of 517 nm. Measurements were also performed on control samples consisting of 1 mL of DPPH solution (27 μg/mL). Scavenging activity percentages—obtained by applying the formula [(A_control_ − A_sample_)/A_control_] × 100—were correlated with EO concentrations. In this way, it was possible to calculate the IC_50_ value, which is a measure of antioxidant activity. Ascorbic acid was used as a positive control. Experiments were conducted in triplicate, and results were expressed as mean of the obtained IC_50_ values ± standard error (SE).

### 4.8. Case Study

#### 4.8.1. Sampling, Samples and Microbial Growth

Sampling was performed on the opposite surfaces of timber (Figure 5a, 85 mm diameter; Figure 5b, 40 × 40 mm) under aseptic conditions, using sterile dry and wet cotton swabs. The samples were transferred to the laboratory at 4 °C. Microbial growth tests for samples were performed in a laminar flow chamber, and manual operations were performed under sterile conditions. The samples were placed on tryptic soy agar (TSA) (Biolife Italiana, Milano, Italy) for the determination of heterotrophic bacteria and on potato dextrose agar (PDA) (Oxoid Limited, Basingstoke, Hampshire, UK, CM139, Thermo Fischer Scientific Inc., Waltham, MA, USA) supplemented with chloramphenicol (0.05 g/L) for the detection of fungi. Petri dishes were incubated at 26 °C and 37°C for 48 and 72 h, respectively.

#### 4.8.2. Isolation and Characterization Procedures

To characterize the isolates, the following media were used: (a) potato dextrose agar (PDA) (Oxoid Limited, CM139); (b) Czapek yeast autolysate agar (CYA agar) (HiMedia, Mumbai, India) (5.0 g, sucrose 30.0 or 200.0 g/L, agar 15.0 g/L) [74]. Morphological analysis of the isolates was performed using a Leica DMI 3000 B microscope equipped with differential interference contrast optics. Laboratory and portable models of optical OM, including the Nikon Eclipse E600 model (Nikon Instruments Europe B.V. Amsterdam, Netherlands) and stereo SM-Zeiss AxioScope (Carl Zeiss Spa, Milan, Italy) microscope connected to high-resolution digital cameras, were adopted. Fungal inoculum was smeared on the surface of Petri dishes containing malt agar 2% (Difco) added with streptomycin sulfate (Sigma-Aldrich, St. Louis, MO, USA) and 100 µg/mL ampicillin (Fisher BioReagents, Monza, Italy). The plates were incubated at 26 °C for 4 days. *S. rolfsii* was identified by observing the macroscopic and microscopic characteristics of the mycelium-cultivated malt agar (Difco) plate medium, followed by amplification of the nuclear ribosomal ITS (internal transcribed spacer) region with the oligonucleotides ITS1-F (5′-CTTGGTCATTTAGAGGAAGTAA-3′), ITS4 (5′-TCCTCCGCTTATTGATATGC-3′) and ITS4-B (5′-CAGGAGACTTGTACACGGTCCAG-3′). The amplified DNA was purified and sequenced [75].

## 5. Conclusions

This study showed significant variability in the EO composition of four *L. angustifolia* L. populations collected at full flowering from different geographical areas. We observed that two pairs of LaPE/LaPS and LaCC/LaPRV EOs, on the basis of both their chemical composition and percent model affinity (PMA) index, were identified.

The in vitro antifungal activity of lavender EOs against *S. rolfsii*, *B. cinerea*, *F. verticillioides*, *P. chrysosporium* and *T. cingulata* showed a wide range of variability responses. Furthermore, the in vitro antioxidant activity by DPPH assay showed variability related to the different compositions of EOs.

If confirmed by further studies, the antifungal activities of lavender EOs for artwork recovery could be useful in setting up an advanced green biological strategy as an alternative to synthetic biocides and toxic compounds.

Lavender oil, which is now used as a flavoring ingredient in food processing, could be used as an antioxidant to preserve foods and also protect artworks from biodeterioration. However, it is advisable to be very careful when handling lavender EOs because of their potential for harm from certain chemical components they contain.

## Figures and Tables

**Figure 1 molecules-28-00392-f001:**
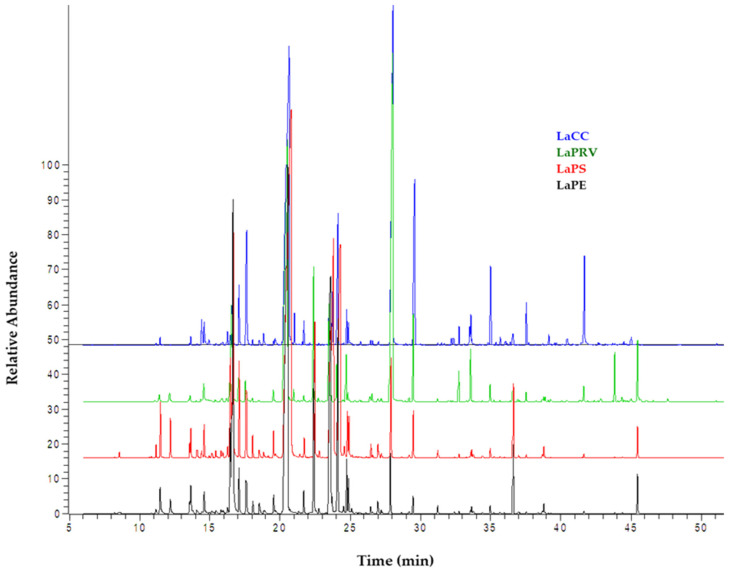
The GC-MS TIC chromatograms of lavender EO samples.

**Figure 2 molecules-28-00392-f002:**
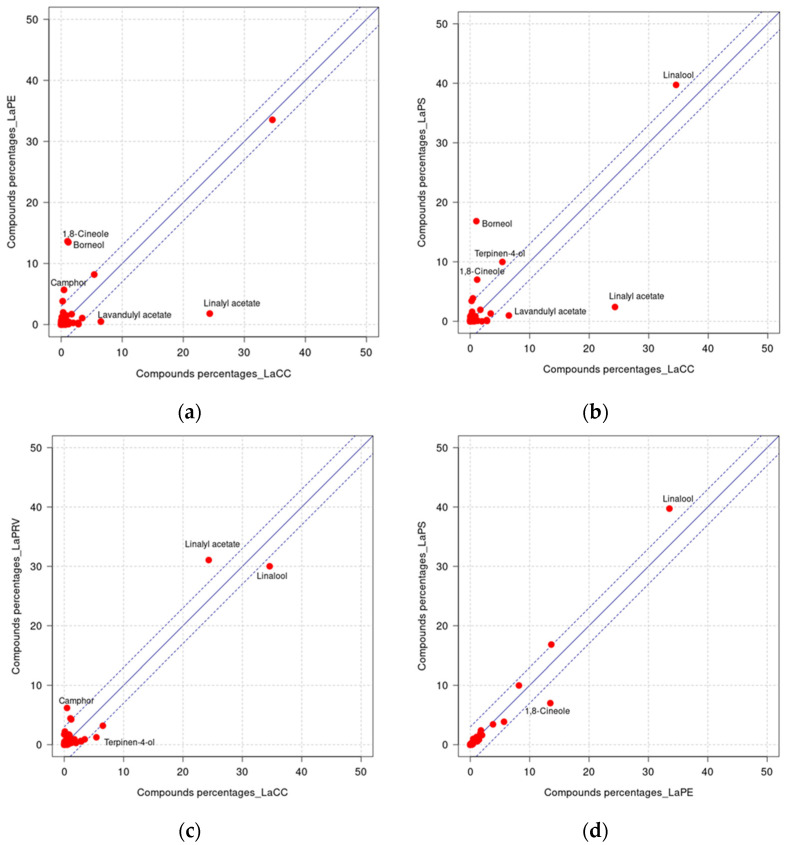
Cross plots between compositions, EO LaX compound percentages vs. EO LaY compound percentages, where X and Y represent different lavenders: LaCC vs. LaPE (**a**); LaCC vs. LaPS (**b**); LaCC vs. LaPRV (**c**); LaPE vs. LaPS (**d**); LaPE vs. LaPRV (**e**); LaPS vs. LaPRV (**f**).

**Figure 3 molecules-28-00392-f003:**
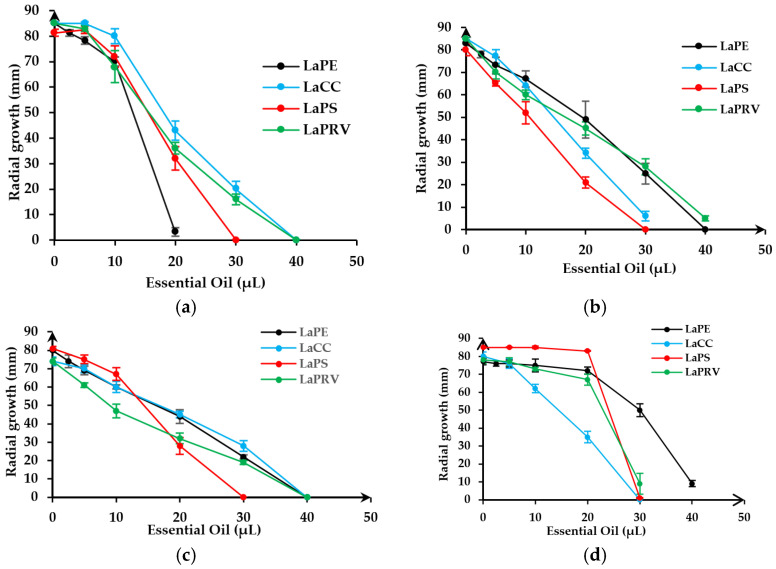
Toxic effect of different concentrations of lavender EOs on the radial growth of *P. chrysosporium* (**a**), *S. rolfsii* (**b**), *B. cinerea* (**c**), *T. cingulata* (**d**) and *F. verticillioides* (**e**). Fungal growth on Thiram has was out as a positive control (**f**). The tests were conducted in triplicate ± SE.

**Figure 4 molecules-28-00392-f004:**
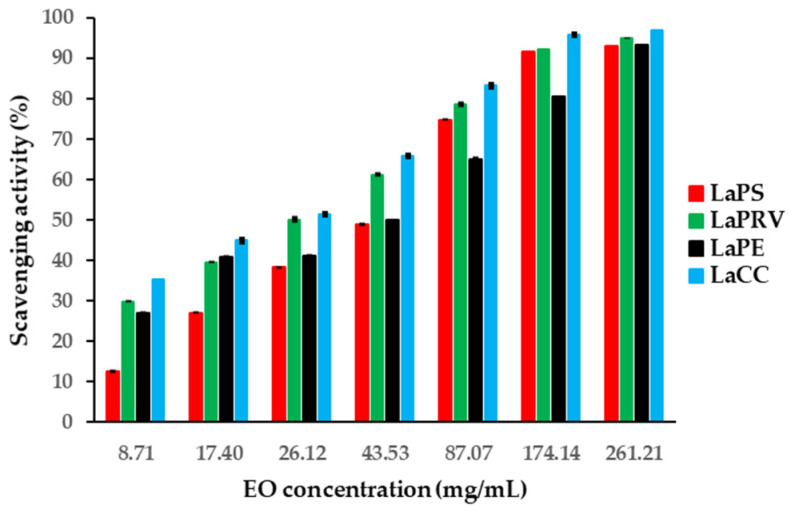
Scavenging effect on four *L. angustifolia* Eos on DPPH assays at different concentrations ranging from 8.71 mg/mL to 261.21 mg/mL. Data are expressed as mean values ± SE (*n* = 3).

**Figure 5 molecules-28-00392-f005:**
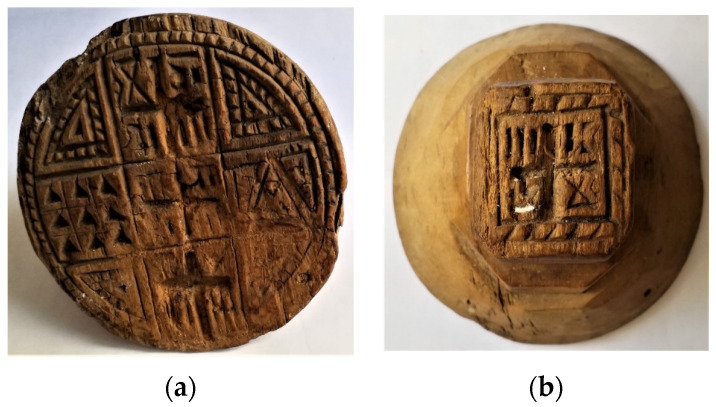
Olive wood stamp for the Offering Bread (**a**,**b**: double-sided) used only at the exchange of greetings for the feast of Holy Easter in the Catholic community of Villa Badessa (Rosciano, Pescara, Italy) of the Greek-Byzantine rite [35]. Handmade and preserved on the wall of the Permanent Historical-Ethnoanthropological Exhibition of Villa Badessa (Rosciano, Pescara, Italy).

**Figure 6 molecules-28-00392-f006:**
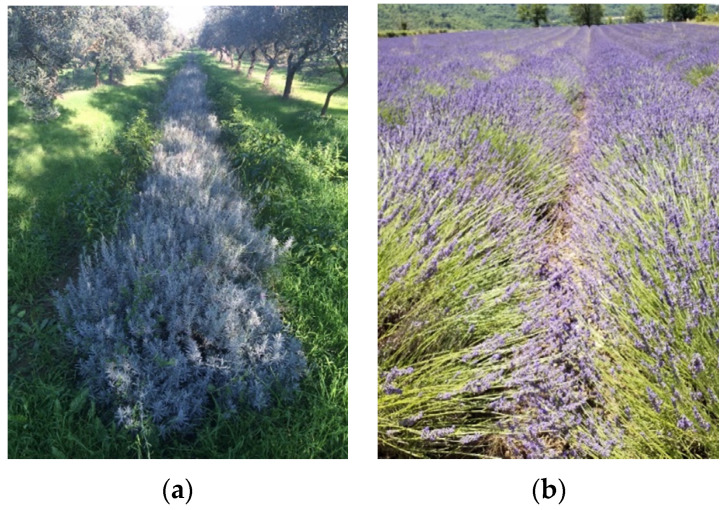
Two examples of the lavender plants sampled in this work. (**a**) LaPE (springtime), Rosciano, Italy, and (**b**) LaPRV, (summertime) Provence Region, France.

**Figure 7 molecules-28-00392-f007:**
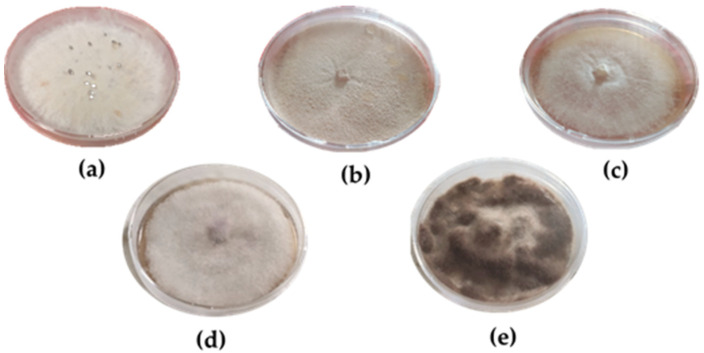
Fungal isolates grown on PDA medium. (**a**) *S. rolfsii*, (**b**) *P. chrysosporium*, (**c**) *T. cingulata*, (**d**) *F. verticillioides* and (**e**) *B. cinerea*.

**Table 1 molecules-28-00392-t001:** Chemical composition of EOs extracted from *L. angustifolia* flowers.

N.	Compound	Exp RI	Ref RI	Area % ± SD LaCC	Area % ± SD LaPE	Area % ± SD LaPS	Area % ± SD LaPRV	Abbr.
1	Tricyclene	922	926	-	0.03 ± 0.01	0.04 ± 0.01	0.02 ± 0.01	BM
2	*α*-Thujene	929	930	0.06 ± 0.01	0.19 ± 0.03	0.25 ± 0.01	0.05 ± 0.01	BM
3	*α*-Pinene	936	939	0.22 ± 0.01	1.26 ± 0.07	1.14 ± 0.02	0.36 ± 0.02	BM
4	Camphene	950	954	0.03 ± 0.01	0.69 ± 0.04	0.81 ± 0.02	0.5 ± 0.01	BM
5	Sabinene	976	975	0.06 ± 0.01	0.47 ± 0.07	0.25 ± 0.05	0.08 ± 0.03	BM
6	*β*-Pinene	977	979	0.22 ± 0.03	1.18 ± 0.01	0.57 ± 0.04	0.25 ± 0.06	BM
7	1-Octen-3-ol	984	979	0.04 ± 0.01	0.16 ± 0.01	0.18 ± 0.01	0.03 ± 0.01	OT
8	3-Octanone	990	983	0.75 ± 0.03	0.08 ± 0.01	0.13 ± 0.01	0.11 ± 0.01	OT
9	Myrcene	993	990	0.78 ± 0.06	0.79 ± 0.02	0.58 ± 0.02	0.96 ± 0.05	AM
10	Butyl butanoate	999	994	0.26 ± 0.03	0.03 ± 0.01	0.03 ± 0.01	0.03 ± 0.01	OT
11	*α*-Phellandrene	1001	1002	-	0.09 ± 0.01	0.08 ± 0.01	0.02 ± 0.01	MM
12	3-Carene	1008	1011	0.06 ± 0.01	0.12 ± 0.01	0.12 ± 0.01	0.07 ± 0.01	BM
13	*α*-Terpinene	1016	1017	0.06 ± 0.02	0.13 ± 0.01	0.14 ± 0.02	0.06 ± 0.01	MM
14	3-Undecen-1-yne	1018	1120	0.13 ± 0.01	0.09 ± 0.02	0.1 ± 0.01	0.14 ± 0.01	OT
15	*p*-Cymene	1025	1024	0.38 ± 0.01	0.19 ± 0.02	0.26 ± 0.02	0.1 ± 0.01	MM
16	Limonene	1030	1029	0.25 ± 0.02	3.83 ± 0.41	3.43 ± 0.14	0.75 ± 0.03	MM
17	1,8-Cineole	1032	1031	1.18 ± 0.08	13.48 ± 0.44	7.0 ± 0.12	4.26 ± 0.1	BMO
18	(*Z*)-*β*-Ocimene	1042	1037	1.72 ± 0.04	1.7 ± 0.27	1.94 ± 0.06	0.91 ± 0.04	AM
19	(*E*)-*β*-Ocimene	1053	1050	3.45 ± 0.05	1.06 ± 0.01	1.31 ± 0.04	0.91 ± 0.02	AM
20	*γ*-Terpinene	1061	1059	0.14 ± 0.02	0.41 ± 0.01	0.43 ± 0.02	0.12 ± 0.01	MM
21	*cis*-Sabinene hydrate	1070	1070	0.19 ± 0.02	0.47 ± 0.04	0.21 ± 0.01	-	BMO
22	U	1076	-	0.46 ± 0.01	0.17 ± 0.09	0.14 ± 0.02	-	-
23	Terpinolene	1089	1088	0.11 ± 0.01	0.63 ± 0.03	0.54 ± 0.01	0.43 ± 0.02	MM
24	Linalool	1107	1096	34.6 ± 0.43	33.54 ± 0.21	39.73 ± 0.43	30.02 ± 0.26	AMO
25	Octen-3-yl acetate	1119	1112	0.73 ± 0.01	-	-	0.35 ± 0.01	OT
26	allo-Ocimene	1133	1132	0.5 ± 0.11	0.34 ± 0.33	0.33 ± 0.09	0.19 ± 0.04	AM
27	*trans*-Pinocarveol	1141	1139	0.07 ± 0.02	0.02 ± 0	-	-	BMO
28	Camphor	1147	1146	0.47 ± 0.01	5.68 ± 0.1	3.87 ± 0.03	6.2 ± 0.03	BMO
29	Borneol	1169	1169	1.05 ± 0.02	13.65 ± 0.18	16.83 ± 0.34	4.43 ± 0.07	BMO
30	Lavandulol	1173	1169	1.98 ± 0.04	0.27 ± 0.02	-	0.29 ± 0.08	AMO
31	Terpinen-4-ol	1180	1177	5.42 ± 0.02	8.2 ± 0.16	9.98 ± 0.06	1.25 ± 0.01	MMO
32	*α*-Terpineol	1192	1188	0.88 ± 0.02	1.48 ± 0.03	0.86 ± 0.06	1.61 ± 0.05	MMO
33	Hexyl butanoate	1194	1192	0.51 ± 0.04	0.85 ± 0.03	0.75 ± 0.07	0.27 ± 0.02	OT
34	Estragole (Metil chavicol)	1198	1198	0.04 ± 0.04	0.06 ± 0.08	0.03 ± 0.03	-	OT
35	Isobornyl formate	1229	1239	0.15 ± 0.05	0.22 ± 0.01	0.28 ± 0.01	0.16 ± 0.03	OT
36	Nerol	1230	1229	0.13 ± 0.01	-	-	0.3 ± 0.01	AMO
37	Hexyl-2-metil butyrate	1241	1236	-	0.31 ± 0.03	0.23 ± 0.01	0.03 ± 0.01	OT
38	Cumin aldehyde	1241	1241	0.14 ± 0.02	0.13 ± 0.01	0.1 ± 0.01	0.08 ± 0.02	MMO
39	Hexyl isovalerate	1246	1244	-	0.08 ± 0.01	0.07 ± 0.01	0.14 ± 0.01	OT
40	Linalyl acetate	1265	1257	24.34 ± 0.34	1.8 ± 0.1	2.41 ± 0.05	31.07 ± 0.13	AMO
41	Bornyl acetate	1289	1288	0.04 ± 0.01	0.06 ± 0.01	0.05 ± 0.01	0.1 ± 0	BMO
42	Lavandulyl acetate	1296	1290	6.51 ± 0.02	0.49 ± 0.01	0.98 ± 0.01	3.2 ± 0.03	AMO
43	Hexyl tiglate	1334	1332	0.07 ± 0.01	0.25 ± 0.01	0.17 ± 0.01	0.1 ± 0	OT
44	Eugenol	1351	1359	0.2 ± 0.02	0.03 ± 0.03	0.01 ± 0.01	0.02 ± 0	OT
45	Neryl acetate	1368	1361	0.45 ± 0.02	0.06 ± 0.01	0.07 ± 0.01	0.96 ± 0.03	AMO
46	Copaene	1377	1376	0.05 ± 0.01	-	-	-	BS
47	Daucene	1380	1381	-	0.01 ± 0	-	0.04 ± 0.01	BS
48	*β*-Bourbonene	1381	1388	0.49 ± 0.01	-	-	-	BS
49	*trans*-Myrtanol acetate	1384	1386	0.78 ± 0.05	0.08 ± 0.01	0.1 ± 0.01	1.72 ± 0.06	MMO
50	Hexyl hexanoate	1386	1383	0.15 ± 0.01	0.19 ± 0.01	0.16 ± 0.02	-	OT
51	7-epi-Sesquithujene	1391	1391	-	0.1 ± 0.01	0.06 ± 0.01	-	MSO
52	Sesquithujene	1406	1405	0.01 ± 0	0.05 ± 0	0.05 ± 0	0.02 ± 0	MSO
53	Longifolene	1408	1407	-	0.01 ± 0	-	-	BSO
54	(*E*)-Caryophyllene	1420	1419	2.78 ± 0.04	0.26 ± 0.01	0.21 ± 0.01	0.63 ± 0.02	BS
55	Linalyl butanoate	1426	1423	-	0.03 ± 0	0.03 ± 0	0.04 ± 0.01	AMO
56	*β*-Copaene	1429	1432	0.07 ± 0.01	-	-	-	BS
57	*trans*-*α*-Bergamotene	1436	1434	0.21 ± 0.01	0.04 ± 0	0.03 ± 0.01	0.04 ± 0.01	MS
58	Aromadendrene	1445	1441	0.06 ± 0	0.03 ± 0.01	0.02 ± 0.01	0.02 ± 0.01	BS
59	epi-*β*-Santalene	1448	1447	0.04 ± 0.01	-	-	-	MS
60	*α*-Humulene	1454	1454	0.07 ± 0.01	0.01 ± 0	-	0.02 ± 0.01	MS
61	(*E*)-*β*-Farnesene	1458	1456	0.33 ± 0.01	1.99 ± 0.04	1.61 ± 0.01	0.33 ± 0.01	AS
62	9-epi-(*E*)-Caryophyllene	1463	1466	-	-	-	0.01 ± 0.01	BS
63	Linalyl isovalerate	1467	1468	-	0.06 ± 0.01	0.06 ± 0.01	0.06 ± 0	ASO
64	Dauca-5,8-diene	1469	1472	-	0.02 ± 0	-	0.03 ± 0.01	BS
65	*γ*-Muurolene	1481	1479	1.21 ± 0.03	0.07 ± 0.01	0.08 ± 0.01	0.32 ± 0	BS
66	*α*-Amorphene	1484	1484	0.07 ± 0.01	-	-	-	BS
67	*trans*-Muurola-4(14),5-diene	1496	1493	0.03 ± 0.01	0.01 ± 0.01	-	-	BS
68	(*E*)-Methyl isoeugenol	1500	1492	-	0.02 ± 0.02	-	-	OT
69	(*Z*)-α-Bisabolene	1509	1507	0.05 ± 0.01	0.08 ± 0.01	0.07 ± 0.01	0.06 ± 0.02	MS
70	Lavandulyl isovalerate	1511	1509	-	0.31 ± 0.01	0.23 ± 0.01	0.17 ± 0.02	ASO
71	*γ*-Cadinene	1511	1513	0.01 ± 0	0.03 ± 0.01	-	0.17 ± 0.01	AS
72	6-methyl-α-Ionone	1520	1521	0.27 ± 0.02	0.01 ± 0.01	-	0.05 ± 0.02	OT
73	*δ*-Cadinene	1524	1523	0.02 ± 0	0.02 ± 0	-	0.1 ± 0.02	BS
74	Spathulenol	1578	1578	-	-	-	0.05 ± 0.01	BSO
75	Caryophyllene oxide	1584	1583	2.84 ± 0.14	0.07 ± 0.01	0.02 ± 0.01	0.55 ± 0.01	BSO
76	Guaiol	1603	1600	-	-	-	0.04 ± 0.01	BSO
77	1,10-di-epi-Cubenol	1615	1619	-	-	-	0.09 ± 0.01	BSO
78	*α*-Muurolol	1643	1646	-	0.04 ± 0	0.02 ± 0	1.73 ± 0.05	BSO
79	Bisabolol oxide B	1657	1658	-	0.04 ± 0.01	0.02 ± 0	0.16 ± 0.02	BSO
80	Helifolenol A	1674	1675	0.23 ± 0.01	-	-	0.07 ± 0.01	BSO
81	*α*-Bisabolol	1686	1685	0.11 ± 0.02	1.23 ± 0.06	0.68 ± 0.02	2.23 ± 0.12	MSO

Abbreviations: AM: aliphatic monoterpenes; MM: monocyclic monoterpenes; BM: bi- and tricyclic monoterpenes; AMO: aliphatic monoterpenoids; MMO: monocyclic monoterpenoids; BMO: bi- and tricyclic monoterpenoids; AS: aliphatic sesquiterpenes; MS: monocyclic sesquiterpenes; BS: bi- and tricyclic sesquiterpenes; ASO: aliphatic sesquiterpenoids; MSO: monocyclic sesquiterpenoids; BSO: bi- and tricyclic sesquiterpenoids, OT: others. SD: standard deviation; Exp. RI: experimental retention index; Ref. RI: literature data; U: unidentified component.

**Table 2 molecules-28-00392-t002:** List of terpenes in the lavender EOs.

Terpenes	Abbreviation	LaCC Area %	LaPE Area %	LaPS Area %	LaPRV Area %
Aliphatic monoterpenes Monocyclic monoterpenes Bi- and tricyclic monoterpenes	AM MM BM	6.45 0.94 0.65	3.89 5.28 3.94	4.16 4.88 3.18	2.97 1.48 1.33
Monoterpenes	M	8.04	13.11	12.22	5.78
Aliphatic monoterpenoids Monocyclic monoterpenoids Bi- and tricyclic monoterpenoids	AMO MMO BMO	68.01 7.22 3.0	36.19 9.89 33.36	42.22 11.04 27.96	66.51 4.66 14.99
Monoterpenoids	MO	78.23	79.44	81.22	86.16
Aliphatic sesqiuterpenes Monocyclic sesquiterpenes Bi- and tricyclic sesquiterpenes	AS MS BS	0.34 0.37 4.78	2.02 0.13 0.42	1.61 0.1 0.31	0.5 0.12 1.15
Sesquiterpes	S	5.49	2.57	2.02	1.77
Aliphatic sesquiterpenoids Monocyclic sesquiterpenoids Bi- and tricyclic sesquiterpenoids	ASO MSO BSO	- 0.12 3.07	0.37 1.38 0.16	0.29 0.79 0.06	0.23 2.25 2.69
Sesquiterpenoids	SO	3.19	1.91	1.14	5.17
Others	OT	3.3	2.38	2.14	1.43

Abbreviations: AM: aliphatic monoterpenes; MM: monocyclic monoterpenes; BM: bi- and tricyclic monoterpenes; AMO: aliphatic monoterpenoids; MMO: monocyclic monoterpenoids; BMO: bi- and tricyclic monoterpenoids; AS: aliphatic sesquiterpenes; MS: monocyclic sesquiterpenes; BS: bi- and tricyclic sesquiterpenes; ASO: aliphatic sesquiterpenoids; MSO: monocyclic sesquiterpenoids; BSO: bi- and tricyclic sesquiterpenoids, OT: others.

**Table 3 molecules-28-00392-t003:** Similarity and diversity in the chemical compositions of lavender EOs from four different geographical provenances, assessed with the PMA similarity index (minimum similarity = 0; max similarity = 1) and the Pielou index (minimum diversity = 0; max diversity = 1), respectively. The respective standard errors over the three technical replicates are reported.

EOs	Similarity				Diversity	
	LaCC	LaPE	LaPS	LaPRV	Shannon Entropy	Pielou Index
LaCC	1	0.535 ± 0.001	0.552 ± 0.001	0.719 ± 0.001	2.376 ± 0.007	0.539 ± 0.002
LaPE		1	0.866 ± 0.001	0.592 ± 0.001	2.463 ± 0.010	0.559 ± 0.002
LaPS			1	0.565 ± 0.001	2.266 ± 0.006	0.514 ± 0.001
LaPRV				1	2.310 ± 0.007	0.524 ± 0.002

**Table 4 molecules-28-00392-t004:** IC_50_ values (mg/mL) and scavenging activity range (%) of the four lavender EOs analyzed.

Sample	IC_50_ (mg/mL)	% Scavenging Activity
LaPS	49.63 ± 0.42	13.08–93.04
LaPE	48.00 ± 0.35	28.40–94.24
LaPRV	33.53 ± 0.23	29.98–94.87
LaCC	26.26 ± 0.21	35.29–98.70

## Data Availability

Not applicable.

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
