# Peer review of "Chemical Profiles, In Vitro Antioxidant and Antifungal Activity of Four Different Lavandula angustifolia L. EOs"

_molecules, 2023, doi:10.3390/molecules28010392_

Round 1

Reviewer 1 Report

The aim of this study is to determine the essential oil composition and also the biological performance of four L. angustifolia which are collected from different regions. The article is well documented and also methods are clarified. But there are some minor issues to correct before publication. There are many italic, and capital letters mistakes throughout the article. You can see this in the uploaded document.

- It should be better to add some comparisions with the other L. angustifolia EOs.

- Generally, it will be better to use more than 1 antioxidant assay to evaluate the antioxidant potential. But in the paper, only one method was used. Could you please explain why only one method was used for determination? And also it would be better to add some comparisons with other articles and add more references. And also it would be better to add some information about the relationship between EOs content and antioxidant effects.

In my opinion, this article has value and is worth publishing after minor corrections.

Best regards.

Author Response

Thank you very much for your valuable suggestions.

The following are all the changes made in the revised manuscript we are sending out (red color in the text)

- It should be better to add some comparisions with the other L. angustifolia EOs.

Yes. Four references (17-20) were added in the introduction and the major components found in Italian lavenders were reported. (lines 73-80)

- Generally, it will be better to use more than 1 antioxidant assay to evaluate the antioxidant potential. But in the paper, only one method was used. Could you please explain why only one method was used for determination? And also it would be better to add some comparisons with other articles and add more references. And also it would be better to add some information about the relationship between EOs content and antioxidant effects.

Yes. It is generally used to do more than one antioxidant assay.

In the text now: “We have chosen the DPPH method because it is a popular, quick, easy, and convenient approach for the measurement of antioxidant properties involving the use of free radicals to assess the potential of substances to act as hydrogen donors or free radical scavengers [44,45].” (line 302-305).

We added three references and added information about the relationship between EOs content and antioxidant effect (lines 313-318).

We have corrected all the notes reported in the uploaded document

Reviewer 2 Report

Thank you for submitting your review paper.

The purpose of this study was to determine the chemical composition and in vitro antifungal activity of EOs isolated from four inflorescences of Lavandula angustifolia L. collected in different geographical areas.
After a careful evaluation of the title of the manuscript " Chemical Profiles, in vitro Antioxidant and Antifungal Activity of four different Lavandula angustifolia L. EOs", you can find below my comments:

in the abstract insert a sentence about the case study.

please specify in the introduction the novelty of the work.

in table 1 it is necessary to perform a statistical analysis between samples by ANOVA, Duncan Test, t-test etc.

please make figures 2 clearer.

in line 184 what does the symbol before 4.6 mean.

please complete the conclusions with more details about the results. also write the conclusions in one paragraph.

the part between 150-160 should be moved to the materials and methods chapter.

Author Response

Thank you for your suggestions and criticisms.

We are submitting the revised manuscript with all corrections in red color.

 in the abstract insert a sentence about the case study.

Yes. In the abstract a new sentence about a case study was added at the end of the text.

please specify in the introduction the novelty of the work.

Yes. In the introduction the novelty of the work was added (Lines 102-104)

in table 1 it is necessary to perform a statistical analysis between samples by ANOVA, Duncan Test, t-test etc.

Yes. We performed a statistical analysis between samples through the ANOVA test and the post hoc Duncan test; results are reported in Table S1 (section of supplementary materials). See lines 124 -127 in the manuscript.

please make figures 2 clearer.

Yes. In this new version of the manuscript, the figures 2 were modified with a detailed description in the text (lines 173 – 180) and in the caption (lines 196 – 198)

in line 184 what does the symbol before 4.6 mean.

The symbol “§” was replaced with the word “section”.

please complete the conclusions with more details about the results. also write the conclusions in one paragraph.

Yes. The conclusions were completed with more details about the results. (Lines 507- 521).

the part between 150-160 should be moved to the materials and methods chapter.

Yes. We moved the part between lines 150-160 to the materials and methods

(Lines 428-443).

Reviewer 3 Report

I have concluded the review of your manuscript. I believe that it is an interesting work providing useful information concerning the usage of natural products and more particularly of the essential oils isolated from aromatic plants as an eco-friendly means for biological applications. The phytochemical analytical part seems solid. The quantitative/qualitatitative analysis, of the isolated essential oils, has been presented clearly enough.  Overall, I believe that the present work that it could proceed as it is for publication.

Author Response

Thank you very much for fully supporting our manuscript.

Round 2

Reviewer 2 Report

i accept in present form